# Hierarchical self-assembly of simple hard polyhedra into complex mesophases

**Rodolfo Subert** [1] **& Marjolein Dijkstra** [1,2] ✉

Nature offers many intriguing examples of hierarchically self-assembled mesophases, such as lamellar, gyroid, hexagonal, and cholesteric phases. These structures are typically believed to emerge from complex, competing enthalpic interactions, as observed in block copolymers and amphiphilic surfactants. Here, using extensive Monte Carlo simulations, we demonstrate that even simple achiral hard particles with distorted tetrahedral shapes and purely excluded-volume interactions can spontaneously self-assemble into a diverse range of mesophases and liquid crystal phases, including the unexpected emergence of chiral structures. We attribute the formation of these phases to geometric frustration in the orientational ordering of neighboring particles, induced by their particle shape. The system resolves this frustration by coupling it with an energetically less favorable elastic deformation mode in the orientational ordering, such as twist or splay. We show that simple shape descriptors, such as anisotropy or biaxiality, predict the self-assembly behavior: rod-like particles stabilize cholesteric and twisted lamellar phases, plate-like particles form biaxial and splay nematic phases with randomly distributed splay domains as well as hexagonal cylindrical phases, while moderately anisotropic particles favor gyroid phases. This framework provides valuable insights for designing mesophases in supramolecular chemistry, liquid crystals, colloid science, and nanoparticle assembly.

Liquid crystals represent a unique state of matter that exhibits properties intermediate between those of conventional liquids and crystalline solids[1,2]. Among the various liquid crystalline phases, the simplest is the nematic phase[3,4], in which anisotropic particles lack positional order but align along a common axis, referred to as the nematic director $\hat{\boldsymbol{n}}$. This orientational alignment is responsible for the remarkable optical and mechanical properties of liquid crystals. When spatial variations in the nematic director field $\hat{\boldsymbol{n}}(\hat{\boldsymbol{r}})$ are permitted, a wide variety of complex, spatially modulated phases arise[5]. A notable example is the cholesteric phase, characterized by a helical arrangement of the nematic director that rotates around a fixed axis with a specific pitch length $p$. Other intriguing phases include the twist-bend nematic phase[6], the splay-bend nematic phase, and the recently discovered splay nematic phase[7]. Modulated phases in liquid crystals can be described using the four fundamental deformation modes[8,9] of the nematic director field—splay, twist, bend, and a recently identified fourth mode $\Delta$, associated with saddle-splay[10]. The saddle-splay term in the free-energy density represents the total divergence of a vector field. By applying Gauss' theorem, its volume integral can be transformed into a surface integral. Consequently, this term—typically disregarded in bulk systems since it does not contribute to the free energy away from boundaries or interfaces[10]—has been primarily investigated in the context of defects[11–13].

The shape of a particle can induce a local deformation in the nematic director field. For instance, a banana-shaped particle induces a bend deformation and a pear-shaped particle a splay deformation. However, none of these four elastic modes can uniformly fill the three-dimensional Euclidean space[14,15]. This geometric frustration, affected

[1]Soft Condensed Matter & Biophysics, Debye Institute for Nanomaterials Science, Utrecht University, Utrecht, The Netherlands. [2]International Institute for Sustainability with Knotted Chiral Meta Matter (WPI-SKCM²), Hiroshima University, Hiroshima, Japan. ✉e-mail: m.dijkstra@uu.nl

by the particle shape, is resolved in one of two ways: either by introducing additional, energetically less favorable deformation modes or by incorporating structural defects, such as disclination lines or domain walls. For example, a chiral particle like a helix transfers its chirality by inducing a local double-twist deformation in the liquid crystal's director field[4]. Since a uniform double-twist configuration cannot fill three-dimensional space, the system resolves this geometric frustration by either forming a cholesteric phase, which combines single-twist deformation with the Δ mode[5], or by stabilizing a blue phase, where regions of double twist are organized into cylinders interspersed with a network of disclination lines. For banana-shaped particles with polar symmetry perpendicular to the local nematic director, a spontaneous bend deformation arises. The geometric frustration caused by the inability of this bend deformation to fully extend in three-dimensional space is resolved by coupling bend with either twist or splay deformations. This leads to the formation of twist-bend or splay-bend nematic phases[16]. Alternatively, the system can stabilize a blue phase III, a disordered network of double-twist cylinders[17]. The stability of these structures is governed by a delicate balance between the free energies of the competing phases. Finally, pear-shaped particles[18,19], with polar symmetry aligned parallel to the local director, induce double-splay deformations. Since double-splay cannot uniformly fill three-dimensional space, several structural scenarios have been proposed, including one-dimensional (1D), two-dimensional (2D), and three-dimensional (3D) arrangements of single- or double-splay domains separated by domain walls. Experimentally, a liquid crystal phase composed of splay domains has been observed[7]. This splay nematic phase is conjectured to consist of a 1D modulated structure of single-splay domains, which is a combination of double-splay and Δ modes, separated by domain walls. However, simulations reveal that pear-shaped particles stabilize smectic textures, including lamellar and gyroid phases[20], while the splay nematic remains elusive.

Particles capable of inducing a local Δ deformation in liquid crystals remain a largely unexplored territory. This is particularly relevant for experimental studies on particles and molecules with similar symmetries[21–23], where the characterization of emergent geometrical structures remains elusive. The challenge arises from the geometric frustration inherent to the Δ mode, which cannot uniformly fill Euclidean space[14,15]. In the nematic regime, this frustration is resolved by coupling the Δ mode with either twist or splay[24], while in smectics, the saddle-splay term–interpreted as twice the Gaussian curvature[10]–may induce layer curvature. Yet, the specific structural pathways, the competition with defects, and the nature of the resulting phases that emerge remain poorly understood.

In this Letter, we investigate the self-assembled structures formed by hard particles with a biaxial splay shape. These particles, first proposed in ref. [24] and illustrated in Fig. 1, exhibit a symmetry corresponding to the Δ mode. Our study uncovers a remarkably rich and diverse phase behavior, encompassing six distinct nematic phases, and three different 3D modulated mesophases. The nematic phases include a uniaxial prolate nematic ($N_+$) phase, a uniaxial oblate nematic ($N_-$) phase, a biaxial nematic ($N_B$) phase, a cholesteric ($N^*$) phase exhibiting spontaneous chiral symmetry breaking of achiral particles, a chiral biaxial ($N_B^*$) phase, and a disordered splay nematic ($N_S^D$) phase characterized by randomly distributed splay domains. The mesophases include a hexagonal columnar ($C$) phase and, more remarkably, phases with negative Gaussian curvature, resembling the lamellar ($L$) and gyroid ($G$) phases commonly observed in diblock copolymers[25] and gemini surfactants[23,26]. These findings not only expand our understanding of the role of the Δ mode in liquid crystal systems but also open new avenues for exploring geometrically frustrated self-assembly[27,28].

## Results

We consider a system of hard, distorted tetrahedral-shaped particles, each composed of four faces, six edges, and four vertices, similar to a regular tetrahedron, as illustrated in Fig. 1a. The geometry of these distorted tetrahedra is described by three distinct length scales: the thickness $T$, representing the length of the shortest edge, the width $W$, corresponding to the edge opposite to $T$, and the length $L$, defined as the distance between the midpoints of the edges of length $W$ and $T$. We note that the edge of length $W$ is orthogonal to the edge of length $T$, as the distorted tetrahedron is composed exclusively of two types of isosceles triangles. As shown in Fig. 1(b), the geometry of these particles promotes the Δ mode of director deformations. This Δ mode is characterized by a complex deformation of the director field: an outward tilt along a direction perpendicular to the long axes of the tetrahedra, an inward tilt along a direction orthogonal to both the outward tilt and the long axes, and twist deformations of opposite handedness along directions at 45° relative to both tilt directions. Since these particles are biaxial, it is natural to describe them by their aspect ratios $L/W$ and $W/T$, and the shape parameter $\Sigma = L/W - W/T$. When $\Sigma > 0$, the particles are rod-like and are expected to stabilize a prolate nematic ($N_+$) phase. Conversely, when $\Sigma < 0$, the particles are plate-like and are expected to stabilize an oblate nematic ($N_-$) phase. The special case of $\Sigma = 0$ corresponds to the dual shape, where biaxial nematic ($N_B$) phases are anticipated to be stabilized. This parametrization can result in significant variations in particle volumes, calculated as $v = (\boldsymbol{a} \cdot \boldsymbol{b} \times \boldsymbol{c})/6$, where $\boldsymbol{a}$, $\boldsymbol{b}$ and $\boldsymbol{c}$ are vectors connecting one vertex to the other three. To facilitate comparisons across different equations of state, we normalize the pressure by the particle volume, i.e., $\beta P v$ with $P$ the pressure, $v$ the particle volume, and $\beta = 1/k_B T$ the inverse temperature. To study the orientational ordering of this system more effectively, a unitary, right-handed local reference frame is assigned to each particle. In this frame, the long particle axis $\hat{\mathbf{l}}$ is aligned with the line connecting the two midpoints of the edges of length $W$ and $T$, the medium particle axis $\hat{\mathbf{w}}$ aligns with the edge of length $W$, and the shortest particle axis $\hat{\mathbf{t}}$ is aligned with the shortest edge of length $T$. This choice ensures that any biaxiality in the system emerges along the third vector $\hat{\mathbf{t}} = \hat{\mathbf{l}} \times \hat{\mathbf{w}}$, thereby simplifying the analysis. To study the self-assembly behavior of these hard distorted tetrahedra, we perform Monte Carlo simulations using the HOOMD-blue package[29] in the $NPT$ ensemble, where the particles are allowed to freely translate and rotate. We start from an isotropic phase at low density, and investigate the phase sequence upon slowly compressing the system to higher pressures. Depending on the state point, the equilibration stage required $10^6$ up to $2 \times 10^7$ steps, consistent with the long density autocorrelation times $\tau_c$ observed at high densities (see SI).

We first conduct simulations with 3000 particles to explore a broad range of aspect ratios with either $L/W$, $W/T$, or $L/T$ of at least 4 to increase the stability regime of the nematic phase to lower densities[30]. Subsequently, we increase the number of particles to 24000 while focusing on a narrower set of particle shapes to assess potential size effects. We summarize our results in the state diagram shown in Fig. 2, where the axes correspond to the particle length-to-width ratio $L/W$ and width-to-thickness ratio $W/T$. The background color represents the shape parameter $\Sigma = L/W - W/T$, with green indicating more rod-like particles and purple indicating more disk-like particles. The sequence of thermodynamic phases, evolving from low to high density beyond the isotropic phase, is indicated by the color of the dots in the diagram. We identify three distinct regions in the state diagram, corresponding to rod-like, plate-like, and moderately anisotropic particles. To further characterize the phase sequences, we expand the system size to 24.000 particles and measure the equation of state, represented by the pressure $P$ as a function of packing fraction $\eta$. In addition, we analyze the nematic order parameter, $S_+$ and $S_-$, which are defined as the largest eigenvalue of the standard $3 \times 3$ nematic order parameter tensor $\mathcal{Q} = \langle \frac{3}{2} \hat{\boldsymbol{a}} \otimes \hat{\boldsymbol{a}} - \frac{1}{2} \mathbb{I} \rangle$, where $\hat{\boldsymbol{a}}$ represents either the long particle axis $\hat{\boldsymbol{l}}$ for $S_+$ or the short particle axis $\hat{\boldsymbol{t}}$ for $S_-$ and where the angular brackets represent the average over all particle orientations in the system. Below, we describe these regions and their associated phase behaviors in detail.

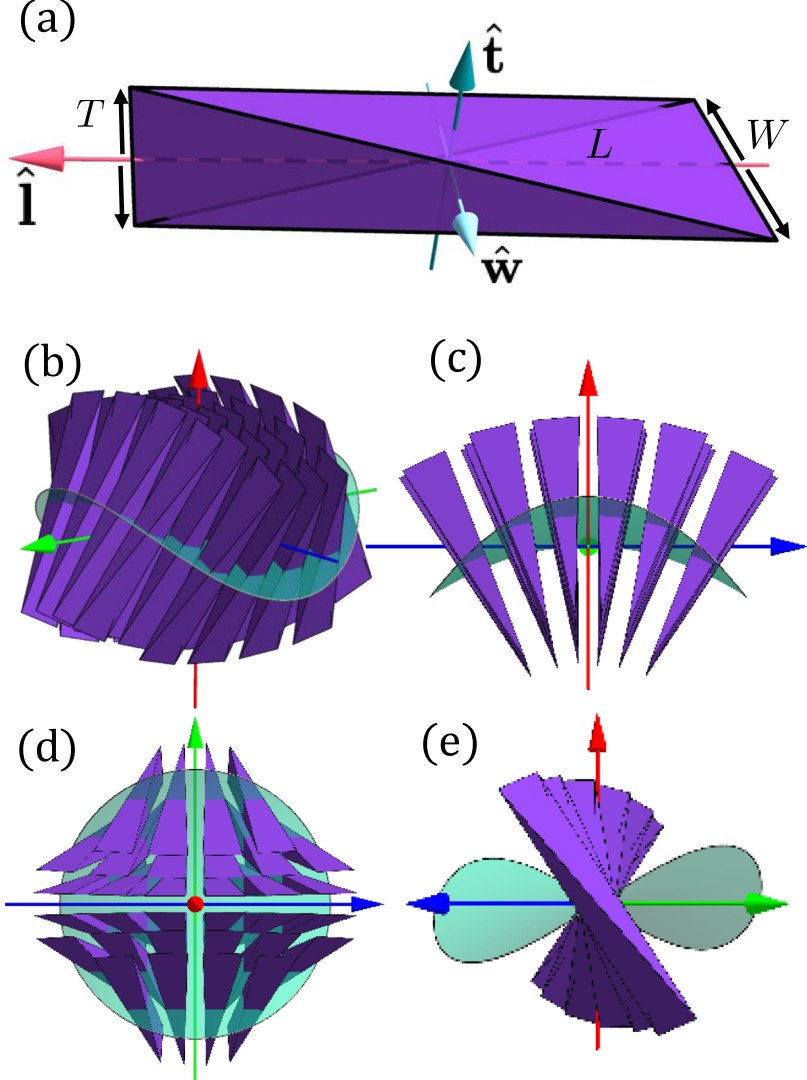

**Fig. 1 | Particle Model of hard, distorted tetrahedral-shaped particles.**
**a** Schematic illustration of a hard, distorted tetrahedral-shaped particle defined by its thickness ($T$), width ($W$), and length ($L$). A unitary, right-handed local reference frame denoted by the colored arrows is assigned to each particle: the long axis $\hat{\mathbf{l}}$ aligns with the midpoint distance between the edges of lengths $W$ and $T$, the medium axis $\hat{\mathbf{w}}$ aligns with the edge of length $W$, and the shortest axis $\hat{\mathbf{t}}$ aligns with the edge of length $T$. **b** The geometry of this particle favors the $\Delta$ mode of director deformations, which is associated with saddle-splay. This $\Delta$ deformation is characterized by an outward tilt of the director field along a direction perpendicular to the long axes of the tetrahedra (**c**), an inward tilt along a direction orthogonal to both the outward tilt and the long axes (**d**), and twists with opposite handedness along directions at 45° relative to both tilt directions (**e**).

## Rod-like particles

We start our investigation by examining the phase behavior of biaxial rod-like particles with aspect ratios $L/W = 5$ and $W/T = 4$, represented by the green dots in Fig. 2a. These particles induce a local $\Delta$ deformation in the director field, a distortion that cannot be uniformly extended into three-dimensional space. We show representative configurations of the various phases observed in Fig. 2b. Remarkably, our simulations reveal that, despite their achiral nature, these particles spontaneously self-assemble from the isotropic ($I$) phase into a chiral nematic ($N^*$) phase via a weakly first-order phase transition. This is reflected by the small jump in the equation of state $\beta P v$ at a packing fraction $\eta = 0.08$ shown in Fig. 3a, and quantified by a nematic order parameter $S_+$ of approximately 0.25, reflecting the orientational averaging over the cylindrical distribution typical of a cholesteric phase, see the supplementary materials (SM) for more details. The relatively low $S_-$ value indicates weak biaxiality, primarily induced by particle shape. Upon further compression, the nematic order parameter value $S_-$ gradually approaches unity, indicating a smooth transition to a biaxial chiral nematic ($N_B^*$) phase, where strong biaxiality is aligned with the chiral axis. At even higher densities, the system transitions into a biaxial chiral smectic phase, also referred to as the twisted lamellar ($L^*$) phase. In this phase, smectic layers form within the plane, consisting of anti-aligned splay domains while maintaining their twisted ordering. The particle centers of mass remain confined within the smectic layers, with their short axes consistently aligned along the cholesteric axis. Figure 3a illustrates how individual splay domains assemble into clusters of up to 3-4 particles (Fig. 3b), which subsequently organize into smectic layers (Fig. 3d, e) oriented along the nematic director. This arrangement preserves a twist along the chiral axis, leading to a tilt in the smectic layering when observed from the side, as depicted in Fig. 3c. Within a cross section perpendicular to the chiral axis (Fig. 3d), the clusters alternate their pointy ends to opposite sides of the smectic layer, forming a characteristic sawtooth pattern. The twisted nature of the smectic phase is

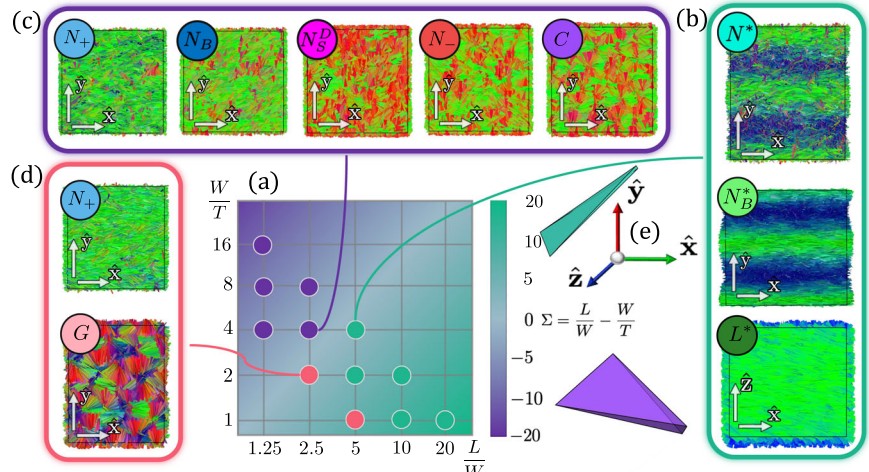

**Fig. 2 | Phase Diagram of hard, distorted tetrahedral-shaped particles. a** State diagram of hard, tetrahedral-shaped particles plotted in the plane of particle length-to-width ratio ($L/W$) versus width-to-thickness ratio ($W/T$). The background color represents the shape parameter $\Sigma = L/W - W/T$ (see color bar), with green indicating more rod-like particles and purple denoting more disk-like particles. An example of a rod-like particle with $L/W = 5$ and $W/T = 5$ is shown in green, and an example plate-like particle with $L/W = 5$ and $W/T = 8$ is shown in purple. The dots represent the simulated particle shapes, and their colors correspond to distinct phase sequences observed as density increases. **b** Representative configurations for rod-like particles with $L/W = 5$ and $W/T = 4$ showing the phase sequence from an isotropic to a chiral nematic $N^*$, biaxial chiral nematic $N_B^*$, and a biaxial chiral smectic or twisted lamellar phase $L^*$. **c** Representative configurations for plate-like particles with $L/W = 2.5$ and $W/T = 4$ progressing from isotropic to a uniaxial prolate nematic phase $N_+$, through a biaxial nematic $N_B$ and a disordered splay nematic $N_S^D$ phase, to a uniaxial oblate nematic $N_-$, and finally to a hexagonal columnar $C$ phase. **d** Representative configurations for moderately anisotropic particles with $L/W = 2.5$ and $W/T = 2$ transitioning from an isotropic to a uniaxial prolate nematic $N_+$, and eventually to a gyroid phase $G$. **e** Particles are colored according to their orientation, following a consistent orientation-based color scheme.

further highlighted by comparing Fig. 3d, e, corresponding to the green and blue regions in Fig. 3c, which shows that the smectic layering at a 1/4 pitch distance is rotated by 90°.

The most striking discovery is the spontaneous breaking of chiral symmetry in a system composed entirely of achiral hard particles. This unexpected phenomenon emerges as a direct consequence of the geometric frustration induced by the locally imposed $\Delta$ deformation. To mitigate this frustration, the system introduces an additional deformation mode. In this case, a single twist provides the most effective solution, ultimately leading to spontaneous chiral symmetry breaking. Since the distorted tetrahedral-shaped particles are achiral, the handedness of the chiral phases is random[4]. This remarkable phase sequence illustrates how achiral rod-like particles resolve geometric frustration from local $\Delta$ mode deformations through twist deformations, leading to the spontaneous formation of intricate chiral structures, including a twisted lamellar phase.

## Plate-like particles

We now turn our attention to plate-like particles, represented by the purple dots in Fig. 2, which exhibit phase behavior distinct from their rod-like counterparts, exhibiting a pronounced tendency toward splay deformations. We show representative configurations of the various phases observed for plate-like particles with aspect ratios $L/W = 2.5$ and $W/T = 4$ in Fig. 2c and supplementary videos. At moderate densities, these particles transition from the isotropic ($I$) phase to a prolate uniaxial nematic ($N_+$) phase. This is quantified in Fig. 3b that reveals a weakly first-order phase transition from the isotropic $I$ phase to the prolate uniaxial nematic $N_+$ phase. This $N_+$ phase is characterized by high nematic order $S_+$ along the long particle axes and low nematic order $S_-$ along the short particle axes. Upon increasing the density, the nematic order parameter along the long axis, $S_+$, reaches its maximum value of 0.7, while disorder persists along the short axis. At a packing fraction of $\eta = 0.18$, a sudden jump in $S_-$ indicates the onset of the biaxial nematic $N_B$ phase, as the short axes rapidly align along a common direction. The $N_B$ phase remains stable within a narrow density

range, where the system exhibits moderate biaxiality with $S_+ = S_- \simeq 0.7$. At $\eta = 0.22$, further increasing the density leads to complete nematic ordering along $\hat{t}$, as indicated by $S_-$ approaching unity, while $S_+$ begins to decrease, marking the emergence of the disordered splay nematic $N_S^D$ phase. This intermediate $N_S^D$ phase is characterized by perfect alignment along $\hat{t}$ and decreasing alignment along $\hat{l}$. During this transition, local clusters with splay perpendicular to $\hat{t}$ and along $\hat{l}$ spontaneously form throughout the system. Eventually, $S_+$ reaches 0.25 at $\eta \sim 0.33$, and the splay angle reaches 90°, resulting in a pure planar or single-splay deformation. This marks the transition to the oblate uniaxial nematic $N_-$ phase, where particles exhibit strong alignment along their short axes. As the density increases further, the $N_-$ phase transitions into a hexagonal columnar ($C$) phase, where particles align their long edges to maximize splay, forming closed circular assemblies composed of tetrahedral particles (Fig. 3f). These circular structures stack vertically (Fig. 3g), creating multiple stacks that organize into a hexagonal lattice of interpenetrating columns (Fig. 3h–j). The particle centers of mass form tubular structures arranged on a hexagonal grid, as clearly visible in Fig. 3j, where the particle size has been reduced to highlight this arrangement. In this geometry, the smectic layers run straight along the tube axes and are perfectly circular in the transverse direction. The interpenetration occurs because the centers of the self-assembled circular structures are thicker than their rims, allowing adjacent columns to interlock. As described in ref. 5, such single-splay deformations can be described as a linear combination of splay and $\Delta$ deformations. This phase resembles the stacked "pizza-like" clusters commonly observed in block copolymers and amphiphilic mesophases.

The most unexpected finding is the system's transition from a prolate $N_+$ to an oblate $N_-$ uniaxial nematic phase. This transition proceeds via an intermediate biaxial nematic $N_B$ phase and a disordered splay nematic $N_S^D$ phase. Notably, this $N_S^D$ phase aligns with the orientation distribution function and the order parameter characteristics predicted for the splay nematic phase derived from the director field ansatz in the literature[31,32], see also See

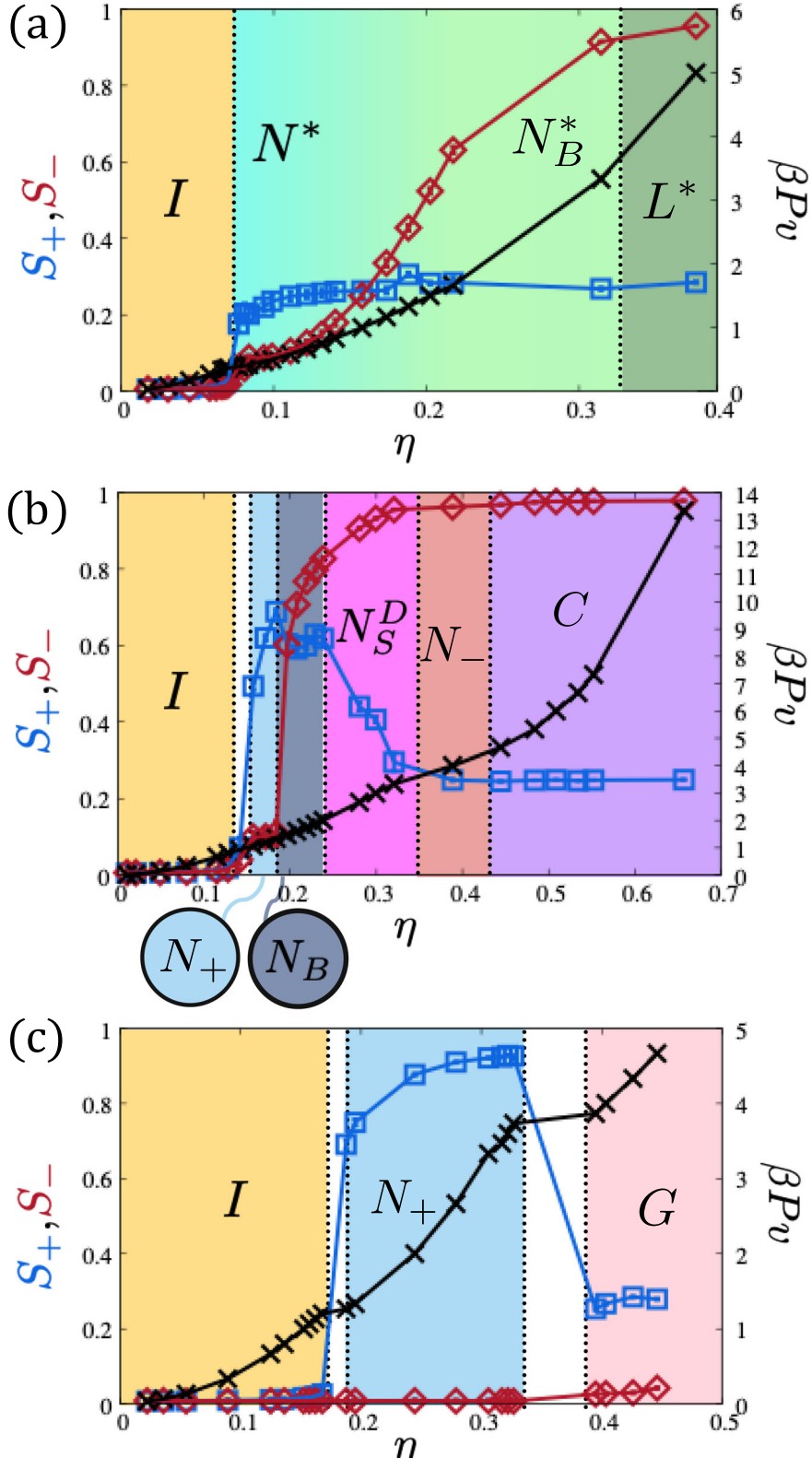

**Fig. 3 | Phase sequences of hard, distorted tetrahedral-shaped particles.** Pressure $\beta P\upsilon$ and nematic orders parameter $S_+$ and $S_-$ as a function of packing fraction $\eta$ for hard, distorted tetrahedral particles. The panels correspond to particles with a length-to-width ratio $L/W$ and width-to-thickness ratio $W/T$ values of $L/W$, $W/T = 5,4$ (**a**); 2.5,4 (**b**); and 2.5,2 (**c**).

Supplementary Material. However, unlike a purely nematic phase, the splay nematic $N_S$ phase typically consists of a one-dimensional modulated structure of single-splay domains separated by domain walls. As these domain walls are likely linked to density modulations, the $N_S$ phase can also be regarded as a smectic phase[33]. In contrast, the $N_S^D$ phase observed in our system remains homogeneous and biaxial, with locally favored splay domains randomly distributed throughout the system.

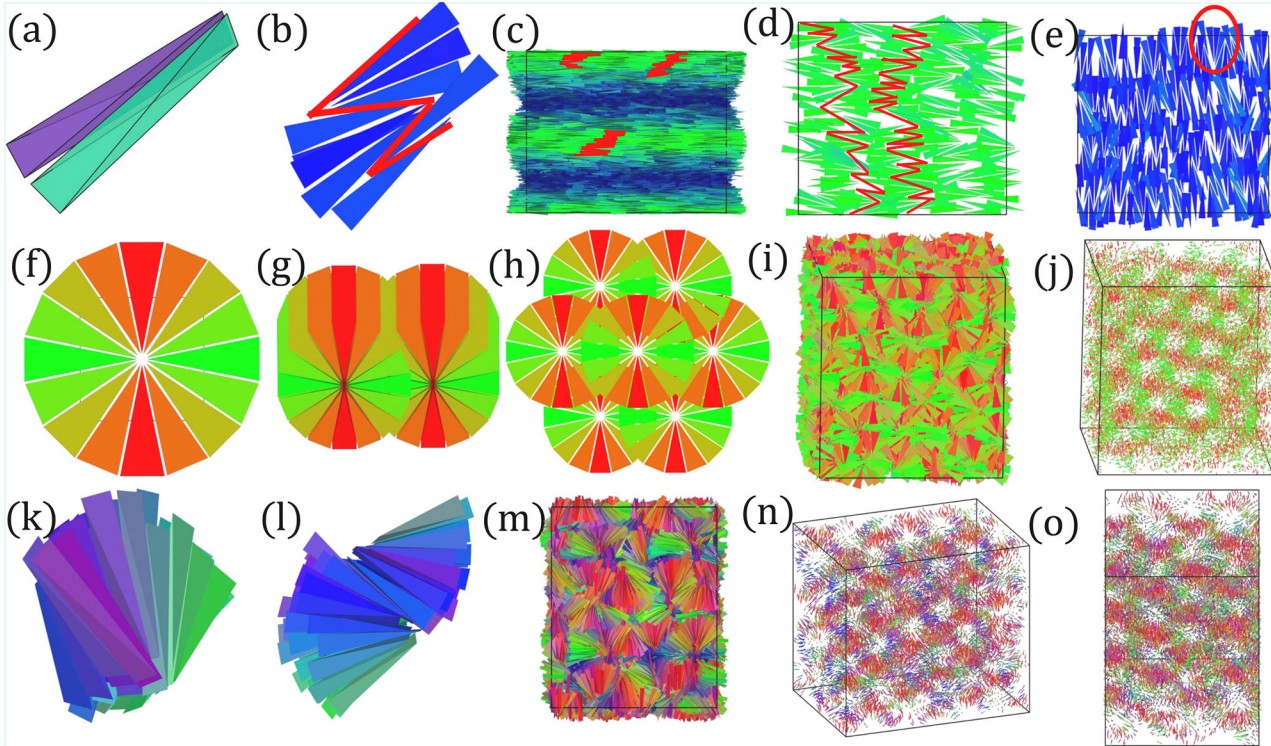

**Fig. 4 | Mesophases of hard, distorted tetrahedral-shaped particles.** All particles are colored according to the orientation scheme shown in Fig. 2e. **a**–**e** Biaxial chiral smectic or twisted lamellar ($L^*$) phase: Schematic of two distorted tetrahedra forming a splay cluster (**a**). Splay clusters of 3-4 particles organize into alternating domains with opposite polarization (**b**). The particle centers of mass remain confined within smectic layers, with their short axes aligned along the cholesteric axis (**c**). These clusters assemble into smectic layers oriented along the nematic director, as observed in simulations (**d**, **e**). Within each layer, alternating particle orientations create a characteristic sawtooth intralayer pattern. The structure preserves a twist along the chiral axis, leading to a tilt of the smectic layers when viewed from the side (**c**). The twisted nature of the layers is further illustrated in (**d**, **e**), corresponding to the green and blue regions in (**c**), where the director has rotated by 90°. **f**–**j** Hexagonal columnar ($C$) phase: Schematic of particles forming closed circular assemblies (**f**), which stack vertically (**g**) to create multiple stacks that organize into a hexagonal lattice of interpenetrating columns (**h**). The interpenetration occurs because the centers of the self-assembled circular structures are thicker than their rims, enabling adjacent columns to interlock. **i** Typical configuration of the hexagonal columnar phase from simulations, replicated in (**j**) with reduced particle size while keeping their centers of mass and orientations fixed. The hexagonal pattern reveals tubular structures arranged on a hexagonal lattice, with smectic layers extending straight along the tube axes and appearing as perfectly circular cross-sections in the transverse direction. (**k**–**o**) Gyroid ($G$) phase: Schematic of a local saddle-splay assembly (**k**) that anti-aligns (**l**) with a neighboring assembly to form a single period modulation. Larger assemblies of saddle-splay domains form a smectic structure with a single gyroid layer, as seen in typical configurations (**m**). This is further illustrated by reducing particle size, which reveals square (**n**) and hexagonal (**o**) patterns at specific viewing angles.

## Moderately anisotropic particles

We now focus our attention on moderately anisotropic tetrahedral-shaped particles with $L/W = 5$ and $W/T = 1$, represented by the magenta dots in Fig. 2. Upon compression from the isotropic phase, these particles assemble either a cholesteric ($N^*$) or a prolate uniaxial nematic $N_+$ phase. A detailed analysis reveals that the prolate nematic phase is thermodynamically favored, emerging from the isotropic $I$ via a weak first-order phase transition (Fig. 3c). This phase exhibits strong orientational order along the particle long axis $S_+$. At higher densities the system undergoes an intriguing transition to a gyroid-like phase. The gyroid is a triply periodic minimal surface exhibiting saddle-splay across the entire surface and partitioning space into two intertwined but non-intersecting chiral networks. Our findings show that this phase is characterized by a single smectic layer defined by the gyroid surface itself, with rods always perpendicular to it while their centers of mass remain anchored to the surface. The modulation arises from alternating local saddle-splay domains induced by the tetrahedral particle shape. The tetrahedral particles first form purely aligned saddle-splay assemblies (Fig. 3k); since these domains cannot continue indefinitely to fill space, their boundaries anti-align with neighboring domains (Fig. 3l) to create a modulation of a single period along one axis. Repetition along all three dimensions ultimately yields the gyroid structure (Fig. 3m). By reducing particle size, this structure reveals hexagonal or square patterns along specific viewing angles (Fig. 3n, o), further emphasizing the triply modulated smectic layer geometry.

Our findings thus show that the single gyroid structure is composed of interconnected, anti-aligned saddle-splay domains separated by domain walls. Surprisingly, despite the presence of these domain walls, our simulations reveal that particles diffuse freely throughout the entire structure. Although one might expect the domain walls to impede particle motion, particles can easily cross them by rotating around their long axes, allowing seamless movement between domains.

## Discussion

In conclusion, we performed extensive Monte Carlo simulations on systems of hard, tetrahedral-shaped particles that induce a local $\Delta$ mode deformation in the nematic director field due to their shape. Since the $\Delta$ mode cannot uniformly fill three-dimensional Euclidean space, the geometric frustration must be resolved either by accommodating a less favorable deformation mode or by incorporating structural defects. We mapped out a state diagram as a function of the width-to-thickness $W/T$ and length-to-width $L/W$ ratios of the particles. Depending on the shape parameter $\Sigma = L/W - W/T$ and the degree of particle anisotropy, we identify three distinct regions in the state diagram, each corresponding to different phase behaviors within the

nematic and smectic regimes. Theoretical predictions from ref. 5 suggest that distorted tetrahedra can induce either a cholesteric phase of arbitrary handedness or a splay nematic phase. Consistent with these predictions, we find that in the nematic regime, the $\Delta$ mode couples to either twist or splay. For moderately anisotropic particles, only uniaxial $N$ nematic phases are stabilized. However, for rod-like particles, characterized with $\Sigma > 0$, the $\Delta$ mode couples to twist, giving rise to the spontaneous formation of uniaxial chiral $N^*$ and biaxial chiral $N_B^*$ nematic phases. In contrast, plate-like particles, with $\Sigma < 0$, exhibit a stronger tendency to splay, transitioning from a uniaxial prolate nematic $N_+$ to a uniaxial oblate nematic $N_-$ phase through an intermediate biaxial nematic $N_B$ and disordered splay nematic $N_S^D$ phase. Although the $N_S^D$ phase observed in our simulations does not fully match the splay nematic structure proposed in ref. 7, it intriguingly displays similar order parameter characteristics. Further experimental and theoretical investigations are required to fully characterize the splay nematic structure.

In our system, two tetrahedra are defined as aligned when their adjacent faces have congruent isosceles triangles facing the same direction, as shown in Fig. 4a. When these triangles face opposite directions, the tetrahedra are considered to be anti-aligned. While aligned tetrahedra promote well-ordered superstructures, anti-alignment introduces defects in the system that play a key role in stabilizing complex mesophases at high densities. In this regime, we find that particle geometry determines the preferred alignment behavior, ultimately determining the stabilized smectic phases. Rod-like particles predominately anti-align along both particle axes, promoting the formation of a biaxial chiral smectic or twisted lamellar $L^*$ phase. Disk-like particles tend to splay along one axis while anti-aligning along the other, resulting in the formation of cylindrical hierarchical structures arranged in a hexagonal lattice, characteristic of the columnar $C^*$ phase. Moderately anisotropic particles splay along both axes, forming saddle-splay-like structures separated by domain walls, which stabilize a single gyroid $G$ phase. These phases closely resemble mesophases found in block copolymers[25] and gemini surfactants[23,26]. Our findings thus reveal a novel connection between liquid crystals and these intricate mesophases.

Furthermore, in the Supplemental Materials, we calculate the twist and splay elastic constants within the framework of Onsager's second-virial theory[34,35]. At the isotropic-nematic transition, we find that the saddle-splay elastic constant $K_{24}$ is consistently negative, showing its prominent role in stabilizing both the cholesteric and disordered splay nematic phase. In addition, the splay elastic constant remains consistently lower than the twist elastic constant ($K_1 < K_2$). This contrasts with the predictions of ref. 5, which suggest that distorted tetrahedra stabilize cholesteric phases when $K_2 < K_1$. This discrepancy may stem from calculating the elastic constants relative to the long particle axis $\hat{\mathbf{l}}$.

Our results clearly demonstrate that hierarchically self-assembled structures can be stabilized purely by entropy, emphasizing the crucial role of geometric constraints in driving complex self-organization. To the best of our knowledge, we are the first to show that hard particles can hierarchically self-assemble into these intricate structures, in contrast to previous studies which focused on simple crystal, liquid crystal, or quasicrystal phases[36–38]. The rich phase behavior observed in our simulations arises solely from entropic effects, with no involvement of enthalpic interactions. This suggests that the formation of these intricate mesophases–similar to those found in block copolymers[25], amphiphiles, and gemini surfactants[23,26]–can be explained by geometric arguments alone. This entropic perspective highlights the importance of hard particle shapes in dictating local ordering and phase behavior via excluded-volume interactions[36–38]. Our findings offer valuable insights for designing novel mesophases in supramolecular chemistry, liquid crystals, colloid science, and nanoparticle assembly, and we hope they inspire further exploration in this exciting field.

## Data availability
Source data files are available with the paper. Further details are available from the corresponding authors upon request. Source data are provided in this paper.

## Code availability
Simulations were performed with an open-source package as referenced in the manuscript. Analysis codes can be made available from the authors upon request.

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

## Acknowledgements

R.S. and M.D. thank Robin Selinger, Jonathan Selinger, Peter Olmsted, Mahesh Mahanthappa and Bela Mulder for useful discussions that helped shaping this work. R.S. acknowledges financial support from the Netherlands Organization for Scientific Research (NWO) (Grant No. OCENW.KLEIN.423). This work was also supported by the European Research Council (Grant No. ERC-2019-ADV-H2020 884902).

## Author contributions

These authors contributed equally: R.S. and M.D. conceived and designed the project, wrote the original draft, discussed, reviewed and edited the manuscript. R.S. performed the numerical simulations and analysis.

## Competing interests

The authors declare no competing interests.
