## [Transparent Peer Review file · Nature Communications]

Hierarchical Self-assembly of Simple Hard Polyhedra into Complex Mesophases

Corresponding Author: Professor Marjolein Dijkstra

Version 0:

Reviewer comments:

Reviewer #1

(Remarks to the Author)

In this article, the authors report very interesting results obtained from extensive Monte Carlo simulations of several self-assembled mesophases of a system consisting of simple achiral hard particles with distorted tetrahedral shapes. Some of the structures found by the authors show evidence that it is possible to obtain spontaneous self-assembly of a wide range of unexpected liquid crystalline mesophases, only by entropic forces.

I consider the work to be very interesting because it finds evidence that the elastic modes (especially the saddle-splay) play an important role in the process to stabilize not common mesophases and the authors report important evidence about how these elastic modes can be found in the bulk, not only on the surface. However, I have some concerns about the article:

1.-The first question is if the authors could make a more extensive description (perhaps in the supplementary section) about which Monte Carlo (MC) movements were used in HooMD to reach the thermodynamic equilibrium for each system. It is important to emphasize in the article that these MC movements allow the system to reach the equilibrium.

2.-One thing that would be very interesting for the reader of this article is to observe that there are not strictly topological defects but the phase transitions due to changes in the values associated with each eigenvalues obtained from the tensors that describe the system, when the density is increased. So it would be possible to show some pictures of isosurfaces from the different scalar order parameters calculated to figure it out possible interfaces, especially for more complex structures like the gyroid phase.

3.-I suppose that simulations were carried out in NPT ensemble. Have you tried to simulate systems in the NVT ensemble, especially for cases where the equation of state has a strong jump, which is evidence of a first-order transition?. The idea is to study a system showing two liquid crystalline mesophases coexisting. Perhaps for future work, it would be great to observe or calculate interfacial properties for this kind of system.

4.-And as minor recommendations, I wonder to see in the supplementary section snapshots of each phase in a larger size. Additionally, the description of the hard model in Figure 1 is not completely clear if the article is printed in b/w. I suggest rotating it a little bit to have a better visualization (similar to the figure reported by Rosseto). On the other hand, I suggest a brief explanation of the color code in the snapshots of mesophases. My question is whether the colors have a meaning or if it is just pictured for visualization. For the case that each color has a meaning, I think it is important to include a color bar.

Reviewer #2

(Remarks to the Author)

The manuscript titled "Hierarchical Self-assembly of Simple Hard Polyhedra into Complex Mesophases" presented a very nice study of the self-assembly behaviour of hard particles with the shape of a distorted tetrahedron using Monte Carlo simulation. A number of different phases, with different shape defining parameters, have been observed, including nematics (both prolate and oblate, chiral and achiral, biaxial, splay nematic), twisted lamellar phase, hexagonal columnar and Gyroid. The results presented is certainly novel and would attract immense interests from researchers in the self-assembly community, and how such intriguing structures can be generated simply from the geometry of the particles would appeal to the general readership of Nature Communication.

I do not have much criticism about the content of the manuscript, and think the simulations have been well executed and results and discussions are convincing. However, I do have some issues mainly to do with the presentation of the data in the figures, which I have listed below and hope these will be dealt with by the authors in the revised version.

1. How the particles are coloured, presumably in some way by orientation, in Figure 2 and 3 is not clearly defined and does not seem to be consistent. Red, green and blue are some of the colours used, but it is not clear what they represent. While in Figure 2b particles align horizontally and parallel to the paper surface seems to be green, in the first panel of Figure 2c green seems to mean the particles are vertical, but moving to Figure 2d vertical particles are now blue. In Figure 3(g) (green horizontal and blue vertical) and (h) (green now becomes vertical) the colour switched direction. A more unified presentation of molecular orientation should be used and clearly explained in the Figure captions.

2. In Figure 3 panels (j), (n) and (o), to show more clearly the structures the particle size has been reduced. Presumably this is by representing a particle at the same centre of mass, but reduce its size proportional in all three dimensions. The fact you can see clearly the hexagonal grid in 3(j) is resulted from the centres of mass of particles are not distributed evenly in space should be clearly explained. This also happens in the gyroid phase, and I just feel there are so much more details that are not described.

3. Furthermore about the formation of the gyroid phase, in the text "single gyroid" is mentioned a couple of times. A gyroid surface divide the space into two subspaces, each consists a "single gyroid" network, and they have opposite chirality. Do the authors mean the particles assemble with one and the same chirality, and only occupy half of the space? This is unlikely as in Figure 3m the whole space are occupied. I suspect what the authors have are double gyroid phase, with two gyroid networks that are mirror images of each other. I would like to see a more detailed description of the phase so it is clearer how the particles arrange exactly to form the phase. I like to find out which part of the molecules would be in the centres of the "single gyroid" subspaces and which goes to the minimum surface. Figure 3n and 3o, even though show the 6- and 4-fold symmetries, unfortunately do not help very much in explaining such details of the structure.

4. In Figure 4, the line colours used, particularly green for S-, is hard to distinguish from the black lines for betaPv, and the fact these lines are presented on top of shaded colours does not help either (there are pink lines on pink background for example). More distinctive colours should be used, and the linewidths should be thicker so that these lines and trends can be better followed.

Reviewer #3

(Remarks to the Author)

The manuscript by Subert and Dijkstra presents a computational study on the phase behavior of hard, distorted tetrahedral-shaped particles. By systematically varying particle shape parameters and packing fraction, the authors uncover a diverse range of liquid crystal mesophases, including the spontaneous emergence of chiral structures. This is an elegant and insightful piece of work that demonstrates how complex phase behavior, including chirality, can arise purely from entropic interactions in systems of achiral particles. I consider this work original and of particular interest to researchers working in soft condensed matter and liquid crystal modeling.

The manuscript is clearly written and well structured, making it easy to follow. I believe it is suitable for publication after addressing the following minor comments:

1. It would be beneficial to provide additional simulation details. For instance, please specify the statistical ensemble used in the Monte Carlo simulations, the types of particle moves implemented, the acceptance ratios, etc. Moreover, it would be helpful to include in the Supporting Information a more detailed explanation of the criteria used to determine when the system has reached equilibrium. For example, Figure 2 in the Supporting Material is claimed to represent equilibrated phases, yet the curves appear to exhibit significant fluctuations.

2. The authors state that their "most striking discovery is the spontaneous breaking of chiral symmetry in a system composed of achiral hard particles." They attribute this behavior to a combination of geometric frustration and saddle-splay deformation, which induces an additional deformation mode. In this context, I wonder whether the distorted tetrahedral particles avoid or escape disclination lines differently, depending on their shape and the topological charge of the defect. Could this lead to the incorporation of additional twist deformations, further contributing to the observed chiral structures?

Version 1:

Reviewer comments:

Reviewer #1

(Remarks to the Author)

The authors not only addressed my comments but are also improving the presentation of the manuscript. I consider the contribution to be ready for publication in Nature Communications

Reviewer #2

(Remarks to the Author)

I am satisfied with the the answers provided by the authors to reviewers' questions, these are very detailed and to the point. The modifications made are accordingly appropriate, and I am happy for the manuscript to be published in its current form.

Reviewer #3

(Remarks to the Author)

The authors have addressed all my previous comments. I consider the manuscript suitable for publication.

Hierarchical Self-assembly of Simple Hard Polyhedra into Complex Mesophases

by Rodolfo Subert and Marjolein Dijkstra

Dear Reviewer #1, Reviewer #2 and Reviewer #3,,

Thank you very much for your reviewing effort and the comments on our manuscript "Hierarchical Self-assembly of Simple Hard Polyhedra into Complex Mesophases" NCOMMS-25-45302.

We appreciated your careful reading, your valuable comments, and your excellent suggestions, which have greatly contributed to the improvement of our work. Below, we address the concerns raised by the reviewers point by point.

Yours Sincerely,

Rodolfo Subert and Marjolein Dijkstra

- **Reviewer #1:**

In this article, the authors report very interesting results obtained from extensive Monte Carlo simulations of several self-assembled mesophases of a system consisting of simple achiral hard particles with distorted tetrahedral shapes. Some of the structures found by the authors show evidence that it is possible to obtain spontaneous self-assembly of a wide range of unexpected liquid crystalline mesophases, only by entropic forces.

I consider the work to be very interesting because it finds evidence that the elastic modes (especially the saddle-splay) play an important role in the process to stabilize not common mesophases and the authors report important evidence about how these elastic modes can be found in the bulk, not only on the surface.

We thank the reviewer for their positive comments.

However, I have some concerns about the article:

1. **The first question is if the authors could make a more extensive description (perhaps in the supplementary section) about which Monte Carlo (MC) movements were used in HooMD to reach the thermodynamic equilibrium for each system. It is important to emphasize in the article that these MC movements allow the system to reach the equilibrium.**

We agree with the reviewer that our original description of the Monte Carlo simulations was not exhaustive. To address this, we have emphasized in the main text the choice of ensemble, total simulation length, and some other key simulation details. Additionally, we introduced Section II in the SI with more simulation details, including the initialization, compression, and equilibration protocols complemented by density autocorrelation functions. These plots show that the correlation time correctly increases with density up to a maximum of 200000 steps in the gyroid phase. These analyses reveal that the choice of 10 million compression steps followed by 1-2 million equilibration steps ensures equilibration, as further confirmed by the steady packing fraction shown in SFig.2, where fluctuations in the packing fraction over a million steps only appear at the fourth decimal place.

We added Section II to the SI and reformulated the paragraph in the main text:

To study the self-assembly behavior of these hard distorted tetrahedra, we perform Monte Carlo simulations using the HOOMD-blue package [29] in the NPT ensemble, where the particles are allowed to freely translate and rotate. We start from an isotropic phase at low density, and investigate the phase sequence upon slowly compressing the system to higher pressures. Depending on the state point, the equilibration stage required 10^6 up to 2×10^7 steps, consistent with the long density autocorrelation times τ_c observed at high densities (see SI).

2. **One thing that would be very interesting for the reader of this article is to observe that there are not strictly topological defects but the phase transitions due to changes in the values associated with each eigenvalues obtained from the tensors that describe the system, when the density is increased. So it would be possible to show some pictures of isosurfaces from the different scalar order parameters calculated to figure it out possible interfaces, especially for more complex structures like the gyroid phase.**

We thank the reviewer for this insightful suggestion. We agree that, regardless of the presence of defects, transitions to both the hexagonal columnar and gyroid phases involve changes in order parameters. This indicates that these transitions can

be described within a Landau–de Gennes framework and suggests the existence of a local observable or order parameter that is spatially modulated according to the particle environment, in contrast to the uniform, space-filling phases such as uniaxial, biaxial, or cholesteric nematics. Furthermore, since particles in smectic phases align perpendicular to the surface defining the layered structure, it is indeed possible to represent smectic layers as isosurfaces and to color them according to local order, highlighting the spatial variation in particle alignment.

In fact, we had previously considered using this approach to improve visualization of the emerging structures, which are not immediately apparent and would benefit from a more intuitive interpretation. We tested several candidate observables, including local density, local curvature, and the eigenvalues of the order parameter tensor, but none yielded clear or unambiguous representations of the complex smectic phases. In practice, thermal fluctuations in the Monte Carlo simulations and the fluid-like nature of the system perturb local density and alignment, preventing the construction of clear isosurfaces. For this reason, we instead present Fig. 3, where configurations are shown with reduced particle size to better illustrate the underlying smectic order.

We acknowledge, however, that this approach may not fully capture the complexity of the emerging phases without a detailed geometrical description. To address this, we have strengthened our description of the gyroid and hexagonal columnar geometries, as reported in our response to Reviewer #2.

3. **I suppose that simulations were carried out in NPT ensemble. Have you tried to simulate systems in the NVT ensemble, especially for cases where the equation of state has a strong jump, which is evidence of a first-order transition?. The idea is to study a system showing two liquid crystalline mesophases coexisting. Perhaps for future work, it would be great to observe or calculate interfacial properties for this kind of system.**

This is indeed an important point. As now clarified in the main text, all simulations were performed in the *NPT* ensemble. We fully agree that investigating phase coexistence and interfacial phenomena—potentially via simulations in elongated boxes in the *NVT* ensemble—would be very interesting, for instance in the study of interfacial properties of liquid crystals. However, addressing these questions is beyond the scope of the present study. Here, our goal was to maintain a clear and linear narrative, focused on demonstrating how simple particles can self-assemble into complex mesophases and establishing a comprehensive phase diagram. We agree that this promising direction is interesting for future work.

4. **And as minor recommendations, I wonder to see in the supplementary section snapshots of each phase in a larger size. Additionally, the description of the hard model in Figure 1 is not completely clear if the article is printed in b/w. I suggest rotating it a little bit to have a better visualization (similar to the figure reported by Rosseto). On the other hand, I suggest a brief explanation of the color code in the snapshots of mesophases. My question is whether the colors have a meaning or if it is just pictured for visualization. For the case that each color has a meaning, I think it is important to include a color bar.**

We thank the reviewer for these helpful suggestions. Given the central role of visualizations in this work, we agree that clarity is crucial, and we have addressed all three of the proposed suggestions.

First, to provide larger snapshots, we have added a dedicated section in the SI. For each phase, we now show orthogonal views of the simulation box along the three axes, together with a perspective view. We believe this substantially improves the interpretability of the structures.

Second, we adjusted the point of view of the particle in Fig. 1a to improve clarity. Although reproducing the exact view of Rosseto *et al.* was not possible, as we also display the particle's reference frame, the revised figure should now be clearer in black-and-white print while retaining the geometrical labeling.

Third, we replotted all configurations in Fig. 4 using a consistent color scheme also aligned with the paper's reference frame. We also added a color legend and updated the captions accordingly to clarify that the colors indicate particle orientation rather than being arbitrarily assigned for visualisation.

• **Reviewer #2:**

The manuscript titled "Hierarchical Self-assembly of Simple Hard Polyhedra into Complex Mesophases" presented a very nice study of the self-assembly behaviour of hard particles with the shape of a distorted tetrahedron using Monte Carlo simulation. A number of different phases, with different shape defining parameters, have been observed, including nematics (both prolate and oblate, chiral and achiral, biaxial, splay nematic), twisted lamellar phase, hexagonal columnar and Gyroid. The results presented is certainly novel and would attract immense interests from researchers in the self-assembly community, and how such intriguing structures can be generated simply from the geometry of the particles would appeal to the general readership of Nature Communication.

We thank the reviewer for their positive comments and for recommending our work for publication in Nature Communications.

I do not have much criticism about the content of the manuscript, and think the simulations have been well executed and results and discussions are convincing. However, I do have some issues mainly to do with the presentation of the data in the figures, which I have listed below and hope these will be dealt with by the authors in the revised version.

- 1. How the particles are coloured, presumably in some way by orientation, in Figure 2 and 3 is not clearly defined and does not seem to be consistent. Red, green and blue are some of the colours used, but it is not clear what they represent. While in Figure 2b particles align horizontally and parallel to the paper surface seems to be green, in the first panel of Figure 2c green seems to mean the particles are vertical, but moving to Figure 2d vertical particles are now blue. In Figure 3(g) (green horizontal and blue vertical) and (h) (green now becomes vertical) the colour switched direction. A more unified presentation of molecular orientation should be used and clearly explained in the Figure captions.**

We thank the reviewer for this careful observation. In a work where visualization plays a central role, we fully agree that a consistent color scheme is essential for interpreting the geometry of the assembled phases.

As the reviewer correctly noted, particles are colored according to their orientations. However, in some cases the simulation boxes were rotated for clarity, which may have given the impression of inconsistencies in coloring. To resolve this, we have added a color legend to Fig. 2 and replotted all configurations using a coherent color scheme that is consistent across the different phases and aligned with the paper's reference frame. The only exception is Fig. 2b (lamellar phase), where we chose to maintain coherence with the cholesteric phases shown in the same panel, rather than strict adherence to the paper's reference frame. This choice is now explicitly explained in the revised caption, which also includes a clear description of the color coding.

The same unified color scheme has also been applied to Fig. 3, with captions updated accordingly. A few exceptions were made for visualization purposes, such as the standalone smectic lamellae in Fig. 3(d–e) and the saddle-splay assemblies in Fig. 3(k–l).

Finally, following a related suggestion from Reviewer #1, we have added a dedicated section to the SI with larger visualizations of all phases, each shown with their corresponding axes and color labels.

We believe these revisions significantly improve the consistency and clarity of the visualizations, thereby enhancing the overall readability of the paper.

- 2. In Figure 3 panels (j), (n) and (o), to show more clearly the structures the particle size has been reduced. Presumably this is by representing a particle at the same centre of mass, but reduce its size proportional in all three dimensions. The fact you can see clearly the hexagonal grid in 3(j) is resulted from the centres of mass of particles are not distributed evenly in space should be clearly explained. This also happens in the gyroid phase, and I just feel there are so much more details that are not described.**

We thank the reviewer for this valuable observation. As correctly noted, in Fig. 3(j), (n), and (o) we present configurations with particle sizes reduced by 80%, while keeping the particle centers of mass and orientations fixed. This representation was chosen to emphasize the smectic character of the spatial particle distribution across curved layers, thereby revealing the underlying geometry of the hexagonal columnar and gyroid phases.

Following this suggestion, we have revised both the caption and the main text to explicitly explain this visualization choice. In particular, we now emphasize how it highlights the geometry of the smectic layers and the relative arrangement of particles therein. Together with the updated captions, these clarifications provide a more complete description of the structural features illustrated in the figures.

In addition, we provide a more detailed description of the various phases in the main text.

Description of the Twisted Lamellar phase

In this phase, smectic layers form within the plane, consisting of anti-aligned splay domains while maintaining their twisted ordering. **The particle centers of mass remain confined within the smectic layers, with their short axes consistently aligned along the cholesteric axis.** Fig. 3(a) illustrates how individual splay domains assemble into **larger** clusters of up to 3-4 particles (Fig. 3(b)), which subsequently organize into smectic layers (Fig. 3(d,e)) oriented along the nematic director. This arrangement preserves a twist along the chiral axis, leading to a tilt in the smectic layering **when observed from the side**, as depicted in Fig. 3(c). **Within a cross section perpendicular to the chiral axis (Fig. 3(d)), the clusters alternate their pointy ends to opposite sides of the smectic layer, forming a characteristic sawtooth pattern.** The twisted nature of the smectic phase is further highlighted **by comparing** Fig. 3(d) and (e), corresponding to the green and blue regions in Fig. 3(c), **respectively**, which shows **that** the smectic layering at **different heights along the chiral director** a $1/4$ pitch distance is rotated by 90° .

Description of the Hexagonal Columnar phase

As the density increases further, the N_c phase transitions into a hexagonal columnar (C) phase, where particles align their long edges to maximize splay, forming closed circular assemblies composed of tetrahedral particles (Fig.3(f)). These circular structures stack vertically (Fig.3(g)), creating multiple stacks that organize into a hexagonal lattice of interpenetrating columns (Fig.3(h-j)). The particle centers of mass form tubular structures arranged on a hexagonal grid, as clearly visible in Fig.3(j), where the particle size has been reduced to highlight this arrangement. In this geometry, the smectic layers run straight along the tube axes and are perfectly circular in the transverse direction. The interpenetration occurs because the centers of the self-assembled circular structures are thicker than their rims, allowing adjacent columns to interlock. As described in Ref.[5], such single-splay deformations can be described as a linear combination of splay and Δ deformations. This phase resembles the stacked "pizza-like" clusters commonly observed in block copolymers and amphiphilic mesophases.

Description of the Gyroid phase

At higher densities the system undergoes an intriguing transition to a gyroid-like phase. The gyroid is a triply periodic minimal surface exhibiting saddle-splay across the entire surface and partitioning space into two intertwined but non-intersecting chiral networks. Our findings show that this phase is characterised by a single smectic layer defined by the gyroid surface itself, with rods always perpendicular to it while their centers of mass remain anchored to the surface. The modulation arises from alternating local saddle-splay domains induced by the tetrahedral particle shape. In this phase, The tetrahedral particles first form purely aligned saddle-splay assemblies (Fig. 3(k)); since these domains cannot continue indefinitely to fill space, their boundaries anti-align with neighbouring domains that (Fig. 3(l)) to create a modulation of a single period along one axis. Repetition along all three dimensions ultimately yields single the gyroid structure (Fig. 3(m)). By reducing particle size, this structure reveals hexagonal or square patterns along specific viewing angles depending on the viewing angle, which becomes clearly visible by reducing the particle size in the simulation configurations as shown in (Fig. 3(n-o)), further emphasizing the triply modulated smectic layer geometry.

Caption of Fig. 3

Mesophases of hard, distorted tetrahedral-shaped particles All particles are coloured according to the orientation scheme shown in Fig. 2(e). (a-e) Biaxial chiral smectic or twisted lamellar (L^*) phase: Schematic of two distorted tetrahedra forming a splay cluster (a). Splay clusters of 3-4 particles organize into alternating domains with opposite polarization (b). The particle centers of mass remain confined within smectic layers, with their short axes aligned along the cholesteric axis (c). These clusters assemble into smectic layers oriented along the nematic director, as observed in simulations (d,e). Within each layer, alternating particle orientations create a characteristic sawtooth intralayer pattern. The structure preserves a twist along the chiral axis, leading to a tilt of the smectic layers when viewed from the side (c). The twisted nature of the layers is further illustrated in (d,e), corresponding to the green and blue regions in (c), where the director has rotated by 90° . (f-j) Hexagonal columnar (C) phase: Schematic of particles forming closed circular assemblies (f), which stack vertically (g) to create multiple stacks that organize into a hexagonal lattice of interpenetrating columns (h). The interpenetration occurs because the centers of the self-assembled circular structures are thicker than their rims, enabling adjacent columns to interlock. (i) Typical configuration of the hexagonal columnar phase from simulations, replicated in (j) with reduced particle size while keeping their centers of mass and orientations fixed. The hexagonal pattern reveals tubular structures arranged on a hexagonal lattice, with smectic layers extending straight along the tube axes and appearing as perfectly circular cross-sections in the transverse direction. showing the hexagonal pattern more clearly by reducing the particle size. (k-o) Gyroid (G) phase: Schematic of a local saddle-splay assembly (k) that anti-aligns (l) with a neighbouring assembly to form a single period modulation. Larger assemblies of saddle-splay domains form a smectic structure with a single gyroid layer, as seen in typical configurations (m). A typical configuration of the gyroid structure from simulations This is further illustrated by reducing particle size, which reveals square (n) and hexagonal (o) patterns at specific viewing angles.

3. Furthermore about the formation of the gyroid phase, in the text "single gyroid" is mentioned a couple of times. A gyroid surface divide the space into two subspaces, each consists a "single gyroid" network, and they have opposite chirality. Do the authors mean the particles assemble with one and the same chirality, and only occupy half of the space? This is unlikely as in Fig. 3(m) the whole space are occupied. I suspect what the authors have are double gyroid phase, with two gyroid networks that are mirror images of each other. I would like to see a more detailed description of the phase so it is clearer how the particles arrange exactly to form the phase. I like to find out which part of the molecules would be in the centres of the "single gyroid" subspaces and which goes to the minimum surface. Figure 3n and 3o, even though show the 6- and 4-fold symmetries, unfortunately do not help very much in explaining such details of the structure.

We thank the reviewer for this careful observation and the opportunity to clarify our terminology. The gyroid is a triply periodic minimal surface, exhibiting saddle-splay across the entire surface, and its emergence in this system was anticipated by design. This surface partitions space into two continuous chiral networks, which are intertwined without intersecting. In the scientific literature, the term gyroid usually refers to the minimal surface itself, whereas in soft matter the terms

single gyroid and double gyroid are commonly used to describe whether one or both of the two volumes separated by the surface are occupied.

In our manuscript, we used the term single gyroid to describe the case where particle centers of mass lie directly on the gyroid minimal surface, with their orientations perpendicular to the local surface normal. We now recognize that this terminology is misleading, since in our simulations the particles do not occupy either of the intertwined domains but instead assemble directly on the gyroid surface itself. To avoid confusion, we have revised the text to use the term gyroid without the qualifiers “single” or “double,” which we believe is the most accurate description in this context.

To make this clearer, we have revised the text as reported in the point above.

We hope this clarification resolves the ambiguity regarding particle arrangement and prevents confusion with the more common domain-filling double gyroid phases.

4. **In Figure 4, the line colours used, particularly green for S-, is hard to distinguish from the black lines for betaPv, and the fact these lines are presented on top of shaded colours does not help either (there are pink lines on pink background for example). More distinctive colours should be used, and the linewidths should be thicker so that these lines and trends can be better followed.**

We thank the reviewer for giving us the opportunity to improve the clarity of Fig. 4. We improved the figure according to the Reviewer’s suggestions.

Fig. 4 has been improved by using thicker lines, with line colors chosen to maximize contrast against the background.

• **Reviewer #3:**

The manuscript by Subert and Dijkstra presents a computational study on the phase behavior of hard, distorted tetrahedral-shaped particles. By systematically varying particle shape parameters and packing fraction, the authors uncover a diverse range of liquid crystal mesophases, including the spontaneous emergence of chiral structures. This is an elegant and insightful piece of work that demonstrates how complex phase behavior, including chirality, can arise purely from entropic interactions in systems of achiral particles. I consider this work original and of particular interest to researchers working in soft condensed matter and liquid crystal modeling.

We thank the reviewer for their positive comments.

The manuscript is clearly written and well structured, making it easy to follow. I believe it is suitable for publication after addressing the following minor comments:

1. **It would be beneficial to provide additional simulation details. For instance, please specify the statistical ensemble used in the Monte Carlo simulations, the types of particle moves implemented, the acceptance ratios, etc. Moreover, it would be helpful to include in the Supporting Information a more detailed explanation of the criteria used to determine when the system has reached equilibrium. For example, Figure 2 in the Supporting Material is claimed to represent equilibrated phases, yet the curves appear to exhibit significant fluctuations.**

We agree with the reviewer that our original description of the Monte Carlo simulations lacked sufficient detail, as also noted by Reviewer #1. We have revised the manuscript accordingly, now specifying the choice of statistical ensemble, total simulation length, the types of particle moves, and including additional details in a new Section II in the SI on initialization, compression, and equilibration protocols, acceptance ratios, complemented by exemplary density autocorrelation functions. These analyses confirm that the correlation time correctly increases with density, reaching a maximum of about 2×10^5 steps in the gyroid phase. Based on this, our protocol of 10^7 compression steps followed by 10^6 – 2×10^6 equilibration steps is sufficient to ensure equilibration, as further confirmed by the steady packing fraction shown in SFig.3, where fluctuations in the packing fraction over a million steps only appear at the fourth decimal place.

However, we respectfully disagree with the reviewer’s interpretation of “large fluctuations” in SFig. 3 of the SI. We consider these fluctuations minor, as they are confined to the fourth decimal place. We therefore regard them as a strong indication of equilibrium behavior, since density fluctuations are intrinsic to equilibrium Monte Carlo simulations in the *NPT* ensemble and correctly reflect the finite, albeit large, system size and thermal fluctuations.

We reformulated the paragraph as follows:

To study the self-assembly behavior of these hard distorted tetrahedra, we perform Monte Carlo simulations using the HOOMD-blue package [29] in the *NPT* ensemble, where the particles are allowed to freely translate and rotate. We start from an isotropic phase at low density, and investigate the phase sequence upon slowly compressing the system to higher pressures. Depending on the state point, the equilibration stage required 10^6 up to 2×10^7 steps, consistent with the long density autocorrelation times τ_c observed at high densities (see SI).

- 2. The authors state that their "most striking discovery is the spontaneous breaking of chiral symmetry in a system composed of achiral hard particles." They attribute this behavior to a combination of geometric frustration and saddle-splay deformation, which induces an additional deformation mode. In this context, I wonder whether the distorted tetrahedral particles avoid or escape disclination lines differently, depending on their shape and the topological charge of the defect. Could this lead to the incorporation of additional twist deformations, further contributing to the observed chiral structures?**

We thank the reviewer for raising this intriguing point. Indeed, the spontaneous chiral symmetry breaking in a system of achiral hard particles is one of the most striking findings of our work. This observation opens many questions regarding the interplay between elastic deformations, geometric frustration, and the emergence of defects.

In our simulations, the distorted tetrahedra are designed to induce local saddle-splay deformations, which cannot uniformly fill three-dimensional space. To relieve this frustration, the system couples to other elastic modes. In the case of chiral symmetry breaking, it couples to twist, ultimately driving the formation of the reported cholesteric phases. In these nematic regimes, we do not observe defects, including stable disclination lines. At higher densities, however, when smectic phases emerge, the system does stabilize defect structures: disclination lines appear at the centers of the cylindrical assemblies in the hexagonal columnar phase, and domain walls occur between the saddle-splay building blocks in the gyroid phase.

Our present analysis does not systematically address how particles interact with disclination lines, nor do we observe different topological charges. The nature of defect lines in cholesteric and biaxial nematics is subtle and has only recently been clarified [Phys. Rev. X 4, 031050]. We anticipate that the distorted tetrahedral particles studied here will raise similar questions. Depending on their rod-like or disk-like anisotropy, these particles may influence the escape of disclination lines, for example favoring twist deformations over splay. For rod-like tetrahedral particles, this could be consistent with the reviewer's suggestion that additional twist may minimize frustration near defects. While a definitive answer is beyond the scope of this work, we expect that our findings will motivate future studies, for instance by exploring defect structures in confined geometries where boundary conditions enforce the stabilization of disclination lines.